# *SAMM50*-*rs2073082*, -*rs738491* and -*rs3761472* Interactions Enhancement of Susceptibility to Non-Alcoholic Fatty Liver Disease

**DOI:** 10.3390/biomedicines11092416

**Published:** 2023-08-29

**Authors:** Jinhan Zhao, Xiaoyi Xu, Xinhuan Wei, Shuang Zhang, Hangfei Xu, Xiaodie Wei, Yang Zhang, Jing Zhang

**Affiliations:** 1The Third Unit, The Department of Hepatology, Beijing Youan Hospital, Capital Medical University, Beijing 100069, China; zhaojinhan@mail.ccmu.edu.cn (J.Z.); xiaoyixu2020@163.com (X.X.); weixinhuan@mail.ccmu.edu.cn (X.W.); zhangshuang@ccmu.edu.cn (S.Z.); xuhangfei@mail.ccmu.edu.cn (H.X.); weixiaodie@aliyun.com (X.W.); 2Beijing Institute of Hepatology, Beijing Youan Hospital, Capital Medical University, Beijing 100069, China; 3Menkuang Hospital, Beijing Jingmei Group General Hospital, Beijing Energy Holding Company Limited, Beijing 102399, China

**Keywords:** nonalcoholic fatty liver disease, *SAMM50*, single nucleotide polymorphism, SNP–SNP interactions, aging

## Abstract

Background and aim: Several studies have identified that three *SAMM50* polymorphisms (*rs2073082*, *rs738491*, *rs3761472*) are associated with an increased risk of non-alcoholic fatty liver disease (NAFLD). However, the clinical significance of the *SAMM50* SNP in relation to NAFLD remains largely unknown. Therefore, we conducted a clinical study and SNP–SNP interaction analysis to further elucidate the effect of the *SAMM50* SNP on the progression of NAFLD in the elderly. Methods: A total of 1053 patients over the age of 65 years were recruited. Liver fat and fibrosis were detected by abdominal ultrasound or FibroScan, respectively. Genomic DNA was extracted and then genotyped by Fluidigm 96.96 Dynamic Array. Multivariable logistic regression was used to evaluate the association between NAFLD and SNP. SNP–SNP interactions were analyzed using generalized multivariate dimensionality reduction (GMDR). Results: The risk of NAFLD was substantially higher in people who carried *SAMM50*-*rs2073082* G and -*rs738491* T alleles (OR, 1.962; 95% CI, 1.448–2.659; *p* < 0.001; OR, 1.532; 95% CI, 1.246–1.884; *p* = 0.021, respectively) compared to noncarriers. Carriers of the *rs738491* T and *rs3761472* G alleles in the cohort showed a significant increase in liver stiffness measurements (LSM). The combination of the three SNPs showed the highest predictive power for NAFLD. The *rs2073082* G allele, *rs738491* T allele and *rs3761472* G carriers had a two-fold higher risk of NAFLD compared to noncarriers. Conclusions: Our research has demonstrated a strong correlation between the genetic polymorphism of *SAMM50* and NAFLD in the elderly, which will contribute to a better understanding of the impact of age and genetics on this condition. Additionally, this study provides a potential predictive model for the early clinical warning of NAFLD.

## 1. Introduction

Nonalcoholic fatty liver disease (NAFLD) is the leading cause of chronic liver disease, affecting up to 1.7 billion individuals worldwide and posing a significant health burden. Notably, China has one of the highest prevalence, morbidity and annual mortality rates associated with NAFLD in Asia [1,2]. NAFLD is caused by a confluence of genetic and environmental factors, with hereditary predisposition accounting for approximately 50% of the risk [3,4]. Genome-wide association studies (GWASs) have identified dozens of genetic variants related to NAFLD over the last decade, such as patatin-like phospholipase domain-containing protein 3 (*PNPLA3*), Glucokinase regulatory protein (*GCKR*) and Membrane Bound O-Acyltransferase Domain Containing 7 (*MBOAT7*) [5,6,7]. This indicates that SNPs play a vital role in the development of NAFLD.

Recently, the *SAMM50* variant was found to be closely associated with NAFLD vulnerability [8,9,10]. The *SAMM50* gene encodes Sam50, which is a kind of β-barrel protein distributed in the mitochondrial membrane [11]. It is also involved in the regulation of mitochondrial morphology, function and scavenging mitochondrial reactive oxygen species (ROS) [12]. Several SNPs in the *SAMM50* gene have also been reported to be associated with an increased risk of NAFLD. Clinical studies have revealed that *SAMM50* variants, including *rs738491* and *rs3761472*, increase susceptibility to NAFLD [12,13]. Subsequent reports have also suggested that *rs2073082* is associated with an increased risk of NAFLD [10]. Furthermore, Kitamoto et al. discovered a significant association between the *rs738491* variant and fibrosis in a Japanese cohort, whereas *rs3761472* did not show a correlation [12]. However, *rs738491* was not found to be associated with fibrosis in a Chinese cohort [14]. The findings are inconsistent and require further investigation. In short, the clinical significance of the *SAMM50* SNP in relation to NAFLD remains largely unknown.

In addition, the trend of global aging is becoming increasingly obvious [15]. The World Health Organization (WHO) estimates that the number of individuals aged 60 and above will reach approximately 840 million by 2025 [16]. Evidence indicates that aging increases the incidence of NAFLD, especially mortality from nonalcoholic steatohepatitis (NASH) [7,17,18,19,20]. Unfortunately, little attention has been paid to NAFLD in older adults. Therefore, we conducted this case-control study to investigate the effects of three variants of *SAMM50* and their interactions on NAFLD in the elderly Chinese population.

In the present study, we evaluated the effects of three variants of *SAMM50* and their interactions on the development of NAFLD. The results demonstrated that all three variants were associated with NAFLD in the elderly. Among them, carriers of the *rs2073082* G allele and *rs738491* T allele were associated with susceptibility to NAFLD, and carriers of *rs738491* T and *rs3761472* G alleles were associated with fibrosis. In addition, GMDR analysis demonstrated that the best model for predicting NAFLD included all three SNPs. Thus, this study provides important insights into genetic factors, which may contribute to a better understanding of the underlying mechanisms involved.

## 2. Materials and Methods

### 2.1. Patients and Methods

Elderly citizens in the Beijing Mentougou community who participated in annual free physical examinations were recruited from 1 November 2020 to 30 September 2021. The study protocol was approved by the Ethics Committee of Beijing You’an Hospital, Capital Medical University (IRB number (2020)-133). The approval date was 28 October 2020. The registration number was ChiCTR 2100043106. All subjects signed informed consent forms.

### 2.2. Patient Selection and Enrollment Criteria

NAFLD was diagnosed according to the 2018 AASLD NAFLD management guidelines (i.e., defined as evidence of hepatic steatosis on abdominal ultrasound) [21]. In addition to a diagnosis of NAFLD, the inclusion criteria included the following reasons: (1) residents in the community; (2) volunteered participating in the annual free physical examination, which was provided by the Beijing government for residents older than 65; (3) signed informed consent. Participants were excluded from analyses for the following reasons: (1) missing data on genetic polymorphisms or other important laboratory parameters or medical history; (2) excessive alcohol consumption (>140 g/week for men and >70 g/week for women); (3) inability to obtain reliable abdominal ultrasound results due to specific reasons, such as intestinal gas interference; (4) malignant tumors, HIV and other serious diseases that may affect the nutritional status or organ function; (5) comorbidity with other liver diseases, such as viral hepatitis and autoimmune hepatitis.

Typically, 10–20 residents took part in the routine examination every morning. If a patient was diagnosed with fatty liver by ultrasound, they were transferred to an isolated room where they were recruited and underwent further testing. Controls were selected from residents who arrived after previous NAFLD patients and did not have fatty liver. Ultimately, 1423 residents were examined, and 1053 residents were recruited into the study (Figure 1).

### 2.3. Data Collection

Baseline information for all participants, including their demographics, anthropometrics, clinical parameters, and comorbidities, was measured. All laboratory tests were performed in the central lab of Menkuang Hospital or Beijing Jingmei Group General Hospital, including serum concentrations of alanine aminotransferase (ALT), aspartate aminotransferase (AST), triglyceride (TG), total cholesterol (TC), high-density lipoprotein (HDL), and low-density lipoprotein (LDL), Blood glucose (GLU) and Glycosylated Hemoglobin, Type A1c (HbA1c). Two trained investigators performed an abdominal ultrasound and determined liver fat content and liver stiffness with the FibroScan 502 touch device (Echosens, Paris, France).

### 2.4. Definitions

Using ultrasonography to identify fatty liver, participants were divided into NAFLD and non-NAFLD groups. Weight (kg) divided by the square of height (m) is known as body mass index (BMI). Waist-to-hip ratio (WHR) was the waist circumstance divided by the hip circumstance. The formula for homeostasis model assessment of insulin resistance (HOMA-IR) is fast insulin (pmol/L) × fast blood sugar (FBS, mmol/L))/22.5. Additionally, the non-invasive liver fibrosis score formula non-alcoholic fatty liver disease fibrosis score (NFS) [22] was calculated according to the following equations:

NFS = −1.675 + 0.037 × age (years) + 0.094 × BMI (kg/m^2^) + 1.13 × impaired fasting blood glucose or diabetes mellitus (yes = 1, no = 0) + 0.99 × (AST/ALT) − 0.013 × PLT (×10^9^/L) − 0.66 × albumin (g/dL)

Comorbid diseases were diagnosed according to international guidelines, including hypertension, type 2 diabetes mellitus (T2DM) and metabolic syndrome (Mets). In brief, hypertension is diagnosed when systolic blood pressure ≥ 130 mmHg, diastolic blood pressure ≥ 85 mmHg or taking antihypertensive drugs. T2DM is diagnosed when FBS ≥ 7.0 mmol/L, or HbA1c ≥ 6.5%, or OGTT 2h blood sugar ≥ 11.1 mmo/L or taking hypoglycemic drugs. According to the Joint Statement Criteria, metabolic syndrome (Mets) was defined as the presence of at least three of the following conditions: (1) increased waist circumference (≥90 cm for men or ≥80 cm for women); (2) elevated triglycerides (≥1.70 mmol/L) or medication for elevated triglycerides; (3) reduced HDL-C (<1.0 mmol/L for men and <1.3 mmol/L for women) or using lipid-lowering drugs; (4) elevated blood pressure (≥130/85 mmHg) or taking antihypertensive drugs; and (5) elevated FBS (≥5.6 mmol/L) or taking hypoglycemic drugs [23]. In this study, LSM ≥ 8.2 kPa was used to predict significant hepatic fibrosis [24,25]. BMI ≥ 28 kg/m^2^ was used to characterize obesity.

### 2.5. Genomic DNA Extracting and Genotyping

Genomic DNA was extracted from the patient’s blood specimens by BGI-Shenzhen, China. The concentration and quality of DNA were determined by spectrophotometry (Nanodrop 2000, Thermo Scientific, Wilmington, DE, USA) and diluted to approximately 50 ng/mL before genotyping. Next, genomic DNA was genotyped with the use of a 96.96 genotyping integrated fluidics circuit with customized SNP-type assays on the Juno^TM^ system (Fluidigm, South San Francisco, CA, USA), and quantification on the Biomark^TM^ (Fluidigm, South San Francisco, CA, USA) in accordance with the manufacturers’ instructions. The data were analyzed using Fluidigm SNP Genotyping Analysis software version 4.5.1 (South San Francisco, CA, USA).

### 2.6. Statistical Analysis

The unpaired Student’s *t*-test, one-way ANOVA, or Mann–Whitney U-test (if the data were not normally distributed) were employed for comparing continuous variables, which were presented as means with standard deviations (SD) or medians (25th and 75th percentiles). Categorical variables were reported as numbers and percentages and subjected to the χ^2^ test or Fisher’s exact test. The χ^2^ test was utilized to determine the Hardy–Weinberg equilibrium. Using the χ^2^ test, differences in alleles and genotype distributions between groups were calculated. By using logistic regression analysis, the relationship between genetic variations and NAFLD was assessed, and the odds ratio (OR) with a 95% confidence interval (CI) was obtained. Open source Java software versions 3.0.2 Multifactor Dimension Reduction (MDR) (http://www.epistasis.org/mdr.html accessed on 4 August 2023) [26] and 0.9 Generalized Multivariate Dimensionality Reduction (GMDR) (http://www.ssg.uab.edu/gmdr/ accessed on 4 August 2023) [27,28] for analyzing SNP–SNP interactions. The statistical analyses were performed using the Statistical Package for Social Sciences (SPSS), version 26.0 (SPSS Inc., Chicago, IL, USA). A *p* value < 0.05 was considered statistically significant.

## 3. Results

### 3.1. Baseline Characteristics of Non-NAFLD Controls and NAFLD Patients

A total of 1053 subjects were included, and 28.7% of the subjects were male. Among these patients, 590 (56.03%) had NAFLD. The demographic and clinical characteristics of the two groups are listed in Table 1. The NAFLD patients had a higher BMI, waist and hip circumferences, serum levels of ALT, AST, HbA1C, HOMA-IR, insulin and serum TG and lower HDL than the control group. Controlled attenuation parameter (CAP) and liver stiffness measurement (LSM) were also significantly higher in the NAFLD group than in the controls. However, no significant differences were observed between the two groups in terms of TC, LDL and GLU.

Regarding the metabolic profiles, among NAFLD cases, 465 subjects (78.81%) had hypertension, 272 subjects (46.10%) had T2DM, 537 subjects (91.02%) had Mets, and 425 subjects (72.03%) were obese. The prevalence showed differences between the NAFLD and non-NAFLD groups (*p* < 0.05). Meanwhile, there were no significant differences in stroke incidence and lipid-lowering agent use between the two groups.

### 3.2. Genotypes and Allele Frequencies of rs2073082, rs738491 and rs3761472 in Non-NAFLD and NAFLD Groups

The distribution of each SNP in the non-NAFLD and NAFLD groups was consistent with the Hardy–Weinberg balance and was representative of the population (*p* > 0.05). As described in Table 2, there were significant differences in the genotype of *rs2073082* between the NAFLD group and the control group. The genotype and allele frequencies of *rs738491* differed between the two groups. However, the distribution of the *rs3761472* genotype and allele frequencies was comparable.

### 3.3. rs2073082, rs738491 and rs3761472 Polymorphism and NAFLD Susceptibility Adjusted by Age, Gender and BMI

The relationship between the three variants of *SAMM50* and NAFLD susceptibility under different genetic models is analyzed in Table 3. There was a noticeable association between *rs2073082* polymorphism and NAFLD susceptibility under the homozygous model (GG vs. AA, OR = 1.638, 95% CI, 1.222–2.196, *p* < 0.001) and recessive model (AG + GG vs. AA, OR = 1.836, 95% CI, 1.392–2.421, *p* < 0.001). Significant association remained after adjusting for confounding factors (adjusted OR, 1.691; 95% CI, 1.235–2.315; *p* < 0.001; adjusted OR, 1.962; 95% CI, 1.448–2.659; *p* < 0.001, respectively). Carriers of the G-allele had a higher risk of NAFLD.

Moreover, the data demonstrated strong links between *rs738491* and the presence of NAFLD under the allelic model (T vs. C, adjusted OR, 1.216; 95% CI, 1.005–1.472; *p* = 0.045), the homozygous model (TT vs. CC, adjusted OR, 1.373; 95% CI, 1.050–1.376; *p* = 0.021) and the recessive model (CT + TT vs. CC, adjusted OR, 1.532; 95% CI, 1.246–1.884; *p* = 0.021), which indicated that the T-allele was the risk allele of NAFLD. However, no significant difference existed between any model of *rs3761472* and NAFLD.

### 3.4. Association of Three SAMM50 Gene Variants with Clinical Features

Since previous studies have indicated that the G-allele in *rs2073082*, T-allele in *rs738491* and G-allele in *rs3761472* all increased susceptibility to NAFLD [10,12,13], we classified the patients as carriers and noncarriers in the whole population. As shown in Table 4, the LSM of the *rs738491* T allele and *rs3761472* G allele carriers was significantly higher than that of noncarriers (*p* = 0.040, *p* = 0.019, respectively). Furthermore, our analysis revealed that the significance of LSM remained even after adjusting for sex, age and BMI (Appendix A). As for *rs3761472*, the TC and LDL levels had significant differences between the two groups, even after adjusting for sex, age, BMI and use of lipid-lowering agents in the logistic regression model (Appendix A). Nevertheless, the levels of ALT, AST, TG and HDL between the carriers and the noncarriers of the three genetic variants were not statistically different. The clinical characteristics were also compared in the whole population according to the genotypes of the three SNPs. After correction for multiple linear regression, LSM indicated a difference between groups (Appendix A). There were no differences between the other indicators.

Similar analyses were also performed in the NAFLD population. No statistical significance was discovered (Appendix A).

### 3.5. Analysis of SNP–SNP Interactions

GMDR was used to analyze the interactions between these SNPs. Table 5 presents a summary of the results obtained from the GMDR analysis for the one- to three-locus models. Among them, a significant three-locus model (*p* = 0.0107) involving *rs2073082*, *rs738491* and *rs3761472* was identified, demonstrating the highest training accuracy (56.97%), testing accuracy (TA, 55.52%) and the best cross-validation consistency (CVC: 10/10). A graphical presentation of the best identified interaction models in NAFLD and control groups is given in Figure 2, which demonstrated the distribution of high and low risk of NAFLD in a three-locus genetic model combination (Figure 2). The dendrogram (Figure 3) showed that *rs2073082* and *rs738491* exhibited antagonism effects on NAFLD susceptibility. Fruchterman-Reingold (Figure 4) revealed that *rs3761472* synergized with both *rs738491* and *rs2073082*, resulting in positive information gain values of 0.21% and 0.04% in NAFLD, while *rs2073082* and *rs738491* were antagonistic with negative IG values (−0.76%). Age, sex and BMI adjustments were made to the models.

Importantly, NAFLD risk increased in parallel with the number of loci even after adjusting for confounders. The adjusted OR of one locus, two locus and three locus models were 1.532 (95% CI: 1.144–2.053), 1.809 (95% CI: 1.147–2.853) and 1.892 (95%CI: 1.196–2.993), respectively. Additionally, *rs2073082* G carriers, *rs738491* T carriers and *rs3761472* G carriers had a two-fold higher risk of NAFLD compared to noncarriers (Table 6).

## 4. Discussion

In the last decade, it has become increasingly clear that genetic markers significantly impact the progression of NAFLD. For example, the variant *PNPLA3 rs738409* is associated with increased liver fat content and fibrosis [29]. *GCKR rs780094* can also cause hepatic steatosis, impair mitochondrial β-oxidation and cause a predisposition to fatty liver-related liver disease [30]. Therefore, the identification of genetic markers for NAFLD can be a valuable tool in clinical management, helping with risk assessment and highlighting potential treatment targets [5,31,32]. In the current study, we conducted clinical research and analyzed the interaction between the novel genetic factor *SAMM50* SNPs (*rs2073082*, *rs738491* and *rs3761472*) to understand their impact on the progression of NAFLD in elderly patients. The major findings of the study are as follows: (i) The carriers of the *rs2073082* G allele and *rs738491* T allele significantly increased susceptibility to NAFLD; (ii) the *rs738491* T allele and *rs3761472* G allele carriers had significantly higher levels of LSM than the noncarriers in the whole population; (iii) We found that the best model for predicting NAFLD in elderly individuals included *rs2073082*, *rs738491* and *rs3761472* using GMDR analysis.

Aging, a major risk factor for increased susceptibility to certain diseases, is characterized by a gradual physiological process that affects all systems of the body, including cells, tissues and organs, which ultimately leads to impaired biological function of the organism [33]. Various endogenous and exogenous factors, such as genetic makeup or dietary habits, also have a significant impact on individualized aging [34]. It has been reported that aging, particularly hepatic age, can promote the development of steatosis, non-alcoholic steatohepatitis (NASH) and hepatocellular carcinoma by reducing the regenerative capacity, biotransformation and increasing inflammation of the liver [34]. In a territory-wide retrospective cohort study, Zhang et al. [35] found that most patients with NAFLD and T2DM developed liver-related events or cirrhosis after age 50, suggesting that aging is an independent and strong predictor of NAFLD. Another study, including a cohort of 182 Japanese patients with biopsy-proven NAFLD, also demonstrated that age was closely related to the pathological progression of liver fibrosis [36]. However, epidemiological data on fatty liver in the elderly population are still much less available compared to middle-aged adults and children. Based on the fact that the global elderly population continues to increase, research on aging and NAFLD may contribute to reducing the burden on healthcare systems.

The *SAMM50* gene and its encoded protein SAM50 are important components of the sorting and assembly machinery complex in the mitochondrial outer membrane, involved in maintaining the stability of mitochondrial DNA, respiratory chain complexes, mitochondrial cristae structure [12,14], and the regulation of mitophagy [37]. It has been reported that SAM50 deficiency triggers membrane remodeling and mitochondrial dysfunction, reducing the ability to clear reactive oxygen species (ROS) and causing lipotoxicity and hepatocyte damage, which further contributes to the development of NAFLD [10]. Polymorphisms in *rs2073082*, *rs738491* and *rs3761472* of the *SAMM50* gene were also found to be significantly associated with susceptibility to NAFLD in middle-aged Japanese and Chinese populations through GWAS and a clinical cohort study, respectively [12,13]. Individuals with the homozygous TT genotype of *rs738491* and the GG genotype of *rs2073082* had a lower expression of the *SAMM50* gene, but only decreased protein levels of *rs738491* TT were found in the livers of patients with NAFLD [10]. The above studies imply that these variants may promote mitochondrial dysfunction by reducing *SAMM50* expression, which further leads to the development of NAFLD.

Our results indicate that the *rs2073082* G allele and *rs738491* T allele variations in the *SAMM50* gene significantly increase susceptibility to NAFLD in an elderly population, which is consistent with previous studies reported by Zuyin et al. in a cohort of Han Chinese adults [10]. Furthermore, another population-based study with an average age of around 41 years showed that the *rs3761472* G allele was associated with an increased susceptibility to non-alcoholic fatty liver disease, but this was not found in our cohort. Furthermore, the current research on the relationship between *SAMM50* genetic polymorphism and liver fibrosis is inconsistent. A biopsy-proven study based on a cohort of Japanese liver fibrosis patients with a mean age of approximately 50 years demonstrated that the *rs738491* gene polymorphism, but not *rs3761472*, was strongly associated with fibrosis progression, whereas *rs738491* was not found to be related to fibrosis in another study of a biopsy-proven Chinese liver fibrosis cohort with a mean age of approximately 40 years [12,14]. Unlike the above, our results showed that both the *rs738491* T allele and *rs3761472* G allele carriers were related to liver fibrosis in the whole aging population but not in the NAFLD group. The conflicting findings between *SAMM50* polymorphism and non-alcoholic fatty liver disease may require further validation through large-scale clinical cohorts and in-depth mechanistic studies. However, our finding of this unique correlation of genetic factor gene polymorphisms for NAFLD in older cohorts suggests that age is an important independent risk factor for genetic susceptibility to NAFLD compared to other factors, such as geography, race and ethnicity.

Aging and many aging-related pathological conditions are closely related to mitochondrial function [38]. Early senescent cells have also been shown to have high ROS levels, dysfunctional mitochondria and shorter telomeres [39]. Moreover, targeted elimination of mitochondria within aging cells has been shown to successfully reverse many features of the aging phenotype, including metabolic disorders [40]. Based on the above reports, our study of *SAMM50* gene polymorphism in the elderly cohort may better reflect the impact of this mitochondria-related protein on NAFLD. In addition, several studies have found that mitochondrial dysfunction can disrupt hepatic lipid homeostasis, which contributes to the pathogenesis of NAFLD [41,42,43]. Specifically, loss of mitochondrial function and subsequent reduction in membrane potential have been reported to disrupt cholesterol homeostasis in macrophages and decrease the efflux of cholesterol to apoA-I [44], while enhancing mitochondrial respiration, as well as ATP production, can increase the expression of the ABCA1 protein and effectively promote cholesterol efflux. The above studies indicate that mitochondria play an important regulatory role in cholesterol metabolism. However, our study found, for the first time, that *rs3761472* G carriers had significantly lower TC and LDL than noncarriers. This finding provides compelling evidence for the existence of a potential molecular regulatory mechanism linking age, *SAMM50* polymorphisms, mitochondria and cholesterol metabolism.

The analysis of the interaction of multiple SNPs can provide more accurate disease risk prediction models and a more comprehensive understanding of genetic factors’ impact on disease, improving disease screening and treatment outcomes in clinical practice. Using the GMDR method, our analysis identified that the three-locus model, comprising *rs2073082*, *rs738491* and *rs3761472,* provided the most accurate prediction for NAFLD. Although the OR for the risk of NAFLD in *rs2073082* G carriers was higher than the OR derived from the three-SNP model, in the screening of the best single locus, *rs738491* (TA 0.5260, *p* 0.3770, CVC 6/10) was superior to *rs2073082* and became the best single locus model. The three-locus model, including *rs2073082*, *rs738491* and *rs3761472* (TA 0.5552, *p* 0.0107, CVC 10/10) outperformed the *rs738491* single-locus model. In addition, in our cohort, there were 938 carriers of the *rs2073082* G allele and 541 with NAFLD. A total of 607 patients had simultaneous mutations at all three loci, of which 351 had NAFLD. Therefore, the inconsistent number of carriers may explain the higher OR of *rs2073082* compared to the three genetic loci. However, further confirmation with larger sample sizes is necessary.

GMDR gives the best prediction model based on the accuracy of training and testing, consistency of symbolic testing and CVC. Therefore, it has a relatively reliable reference value. Additionally, *rs2073082* G allele, *rs738491* T allele and *rs3761472* G carriers have a two-fold higher risk of NAFLD compared to noncarriers. Hence, even though *rs3761472* showed no connection when examined separately, we assumed that the interplay of the three variations controlled the development of NAFLD. Chen et al. reported a possible synergistic association between *rs738491*, *rs2143571* and *rs3761472* of the *SAMM50* gene and NAFLD in a middle-aged population [13]. The TA of the three-locus model was 60.79% and CVC was 10/10. However, the best model in their study was a two-locus model combining *rs2143571* and *rs3761472*, with the highest TA (62.21%) and CVC (10/10). We conducted a similar study in an elderly population but included a new SNP-*rs2073082.* Our study found that the three-locus model involving *rs2073082*, *rs738491* and *rs3761472* showed the highest TA (55.52%) and perfect CVC (10/10). Differences in the age of the cohort and inclusion of SNPs may account for the differences in the results of the studies. The genetic marker model discovered using GMDR may help clinical identification of potential NAFLD patients, which is expected to become a useful tool for clinical management and lay a foundation for further mechanistic research.

Our study’s strength was that it was the first to use a geriatric NAFLD cohort for genetic analysis. Moreover, our study focused on investigating the interaction between three SNPs (*rs2073082*, *rs738491* and *rs3761472*) and NAFLD in an elderly population. Furthermore, LSM assists in evaluating the relationship between SNPs and liver fibrosis. Nevertheless, there were still some limitations. First, although liver biopsy is considered the gold standard for NAFLD diagnosis, our cohort was built based on a population with a healthy physical examination, and we can only choose B-ultrasound as the first-line diagnostic method. Second, all participants were recruited from the same city in China, and further research work is required to generalize our results to different ethnic groups and the general population in the future.

## 5. Conclusions

Together, the three SNPs in the *SAMM50* gene provided the most accurate prediction of the predisposition for elderly NAFLD. Among them, the *rs2073082* and *rs738491* genetic variants contributed to NAFLD susceptibility, whereas the *rs738491* T allele and the *rs3761472* G allele were linked to fibrosis. Our research uncovered a novel genetic risk factor for elderly NAFLD, which may help to better understand the mechanism.

## Figures and Tables

**Figure 1 biomedicines-11-02416-f001:**
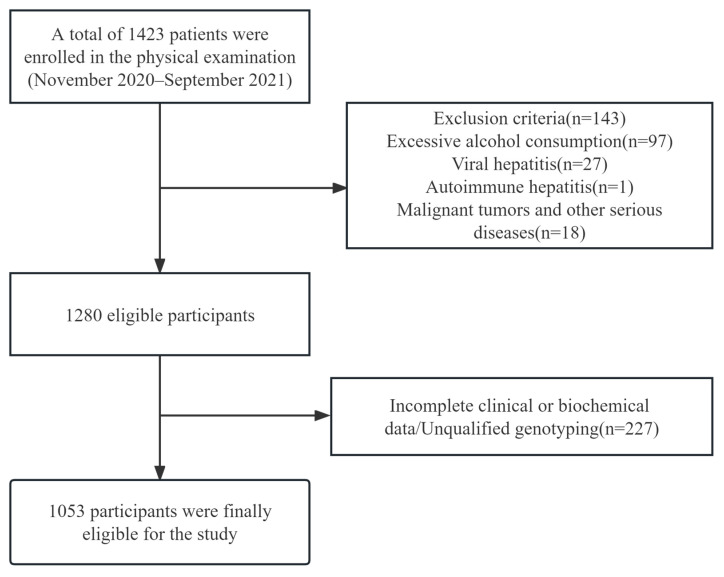
Flow chart of patients’ selection. A total of 1423 patients were initially collected in the cohort, 143 were excluded according to the exclusion criteria, and after exclusion of 227 patients with incomplete data or unqualified genotyping, 1053 patients were finally enrolled in the study.

**Figure 2 biomedicines-11-02416-f002:**
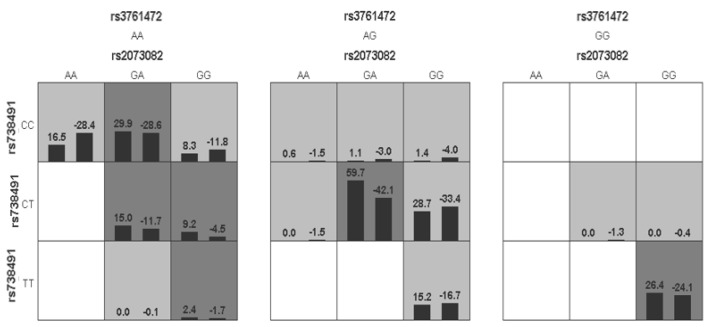
SNP−SNP interactions among *SAMM50 rs2073082*, *rs738491* and *rs3761472* loci in NAFLD and Control subjects. Inside each square, the left bar represents NAFLD subjects (positive score), and the right bar represents control subjects (negative score). The number at the top of each bar is the scoring statistic, which is the product of the affiliation coefficient and the residuals. The scoring statistic categorizes data into high and low risk by calculating whether an individual’s scoring mean exceeds a set threshold (e.g., ≥1). High-risk combinations of genotypes are indicated in dark squares; low-risk genotype combinations are indicated in gray squares; empty squares present the absence of identified genotype combination.

**Figure 3 biomedicines-11-02416-f003:**
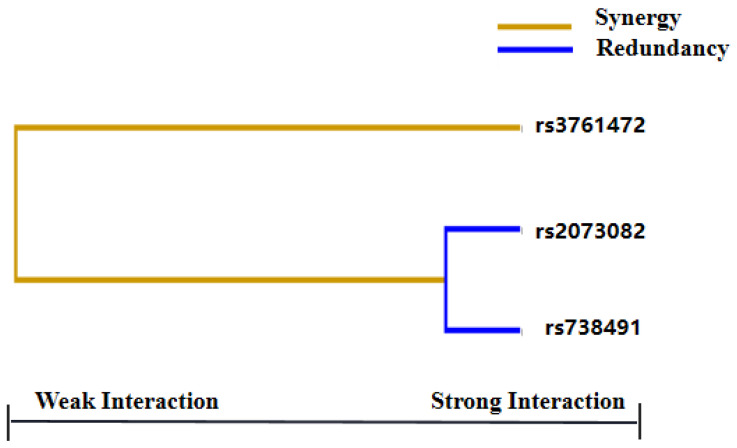
SNP−SNP Interaction Dendrogram. Different types of SNP–SNP effects on NAFLD risk. Orange (synergy); Blue (redundancy or antagonism). Short lines represent stronger interactions, and long lines represent weaker interactions.

**Figure 4 biomedicines-11-02416-f004:**
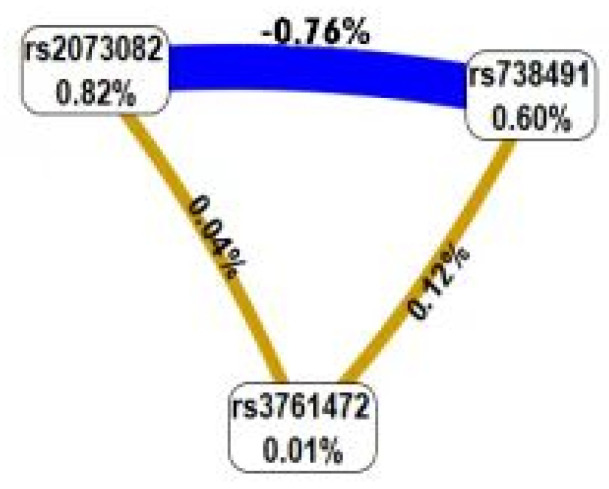
Fruchterman−Rheingold. This interaction model describes the percent of the entropy that is explained by each factor. Each SNP is shown in a box with the percent of entropy below the label. Interactions between SNPs are depicted as lines in different colors. Synergy is depicted as an orange line between SNPs accompanied by a positive percent of entropy, while redundancy is indicated as a blue line accompanied by a negative percent of entropy. *Rs3761472* synergized with both *rs738491* and *rs2073082*, resulting in positive information gain values of 0.12% and 0.04% in NAFLD, while *rs2073082* and *rs738491* were antagonistic with negative IG values (−0.76%).

**Table 1 biomedicines-11-02416-t001:** Clinical characteristics of non-NAFLD controls and NAFLD patients.

	Non-NAFLD (*N* = 463)	NAFLD (*N* = 590)	*p*-Value
Age (year)	70.00 [67.00~75.00]	69.00 [66.75~73.00]	<0.001
Male, *n* (%)	154 (33.26)	149 (25.25)	0.004
Weight (kg)	62.39 ± 9.09	68.31 ± 10.32	<0.001
BMI (kg/m^2^)	24.44 ± 2.77	26.89 ± 3.06	<0.001
Waist circumference (cm)	85.75 ± 7.69	91.12 ± 7.91	<0.001
Hip circumference (cm)	96.07 ± 7.02	100.33 ± 7.16	<0.001
WHR	0.89 ± 0.05	0.91 ± 0.05	<0.001
TBIL (μmol/L)	15.24 ± 5.50	15.65 ± 7.47	0.322
TG (mmol/L)	1.33 ± 0.80	1.78 ± 1.16	<0.001
TC (mmol/L)	4.82 ± 1.17	4.82 ± 1.27	0.939
HDL (mmol/L)	1.21 ± 0.28	1.12 ± 0.22	<0.001
LDL (mmol/L)	3.34 ± 1.07	3.34 ± 1.16	0.980
ALT (U/L)	17.54 ± 7.64	21.47 ± 10.14	<0.001
AST (U/L)	17.40 ± 5.62	19.36 ± 8.57	<0.001
GLU (mmol/L)	6.93 ± 3.82	7.27 ± 2.46	0.078
HbA1C (%)	6.36 ± 1.21	6.70 ± 1.32	<0.001
Insulin	7.80 [5.49~11.05]	10.96 [7.59~15.16]	<0.001
HOMA-IR	2.18 [1.48~3.40]	3.31 [2.25~4.91]	<0.001
CAP (dB/m)	242.93 ± 43.24	295.22 ± 41.11	<0.001
LSM (kPa)	4.51 ± 1.77	5.44 ± 1.99	<0.001
NFS	−0.79 [−1.53~0.03]	−0.73 [−1.40~0.05]	0.870
Hypertension, *n* (%)	345 (74.51)	465 (78.81)	0.035
Mets, *n* (%)	356 (76.89)	537 (91.02)	<0.001
T2DM, *n* (%)	169 (36.50)	272 (46.10)	0.002
Obesity, *n* (%)	45 (0.09)	202 (34.24)	<0.001
Lipid lowering agent, *n* (%)	164 (35.42)	191 (32.37)	0.355
Stroke, *n* (%)	73 (15.77)	92 (15.59)	0.981

Continuous variables are shown as mean ± standard deviation or median [interquartile range]. Categorical values are shown as *n* (%). *p* values were derived from Student’s *t*-test, Mann–Whitney U test or Chi-square test. Abbreviations: BMI, body mass index; WHR, Waist-to-hip ratio; TBIL, total bilirubin; TG, total triglyceride; TC, total cholesterol; HDL, high-density lipoprotein; LDL, low-density lipoprotein; ALT, alanine aminotransferase; AST, aspartate aminotransferase; GLU, Glucose; HbA1C, Glycosylated Hemoglobin Type A1C; HOMA-IR, homoeostatic model assessment of insulin resistance; CAP, controlled attenuated parameter controlled; LSM, liver stiffness measurement; NFS, NAFLD Fibrosis Score; Mets, metabolic syndrome; T2DM, type 2 diabetes.

**Table 2 biomedicines-11-02416-t002:** Distribution of genotypes and allele frequencies of three genetic variants in *SAMM50* in subjects.

Genotypes and Allele	Non-NAFLD	NAFLD	χ^2^	*p*-Value
*SAMM50*-*rs2073082*(G > A)				
Genotypes			12.090	0.002
GG	199 (42.98%)	242 (41.02%)		
GA	198 (42.76%)	299 (50.68%)		
AA	66 (14.25%)	49 (8.31%)		
Alleles			0.912	0.340
G	596(64.36%)	783(66.36%)		
A	330(35.64%)	397(33.64%)		
*SAMM50*-*rs738491*(C > T)			8.722	0.013
CC	164 (35.42%)	161 (27.29%)		
CT	209 (45.14%)	312 (52.88%)		
TT	90 (19.44%)	117 (19.83%)		
Alleles			3.819	0.05
C	537(58.00%)	634(53.70%)		
T	389(42.00%)	546(46.30%)		
*SAMM50*-*rs3761472*(A > G)			0.118	0.943
AA	185 (39.96%)	230 (38.98%)		
AG	225 (48.60%)	290 (49.15%)		
GG	53 (11.45%)	70 (11.86%)		
Alleles			0.109	0.742
A	595(64.30%)	750(63.60%)		
G	331(35.70%)	430(36.40%)		

**Table 3 biomedicines-11-02416-t003:** Study group odds ratios for NAFLD according to genotypes of *SAMM50* single nucleotide polymorphisms in the study group.

Genetic Model	Unadjusted OR, 95% CI	*p*-Value	Adjusted OR, 95% CI ^a^	*p*-Value
*rs2073082*				
G vs. A	0.916 (0.764–1.097)	0.340	1.109 (0.909–1.353)	0.309
GG vs. AA	1.638 (1.222–2.196)	0.001	1.691 (1.235–2.315)	0.001
GG vs. AA + AG	1.084 (0.911–1.290)	0.365	1.084 (0.896–1.313)	0.407
AG + GG vs. AA	1.836 (1.392–2.421)	<0.001	1.962 (1.448–2.659)	<0.001
*rs738491*				
T vs. C	1.189 (0.999–1.414)	0.051	1.216 (1.005–1.472)	0.045
TT vs. CC	1.324 (1.034–1.697)	0.026	1.373 (1.050–1.376)	0.021
TT vs. CC + CT	0.975 (0.785–1.212)	0.822	0.960 (0.757–1.217)	0.737
CT + TT vs. CC	1.462 (1.214–1.760)	<0.001	1.532 (1.246–1.884)	<0.001
*rs3761472*				
G vs. A	1.031 (0.862–1.233)	0.742	1.016 (0.867–1.285)	0.588
GG vs. AA	1.062 (0.797–1.416)	0.680	1.093 (0.800–1.493)	0.575
GG vs. AA + AG	0.970 (0.734–1.256)	0.767	0.975 (0.728–1.305)	0.863
AG + GG vs. AA	1.042 (0.874–1.242)	0.650	1.099 (0.905–1.335)	0.340

^a^ Multiple logistic regression model was adjusted for age, gender, and body mass index.

**Table 4 biomedicines-11-02416-t004:** Comparison of clinical characteristics according to *SAMM50* genotypes within the whole population (Carriers vs. Noncarriers).

	*rs2073082*			*rs738491*			*rs3761472*		
	Carriers (GG + AG)	Noncarriers (AA)	*p*-Value	Carriers (TT + CT)	Noncarriers (CC)	*p*-Value	Carriers (GG + AG)	Noncarriers (AA)	*p*-Value
N	938	115		728	325		638	415	
Age (year)	69.00 [67.00~73.00]	70.00 [67.00~75.00]	0.036	69.00 [67.00~73.00]	70.00 [67.00~75.00]	<0.001	69.00 [67.00~73.00]	69.00 [67.00~74.00]	0.071
Male, *n* (%)	268 (28.57)	35 (30.43)	0.677	205 (28.16)	98 (30.15)	0.529	180 (28.21)	123 (29.64)	0.618
Weight (kg)	65.69 ± 10.24	65.82 ± 10.14	0.898	65.61 ± 10.10	65.93 ± 10.51	0.637	65.29 ± 10.08	66.34 ± 10.42	0.103
BMI (kg/m^2^)	25.83 ± 3.20	25.70 ± 2.98	0.682	25.80 ± 3.14	25.84 ± 3.27	0.863	25.70 ± 3.12	25.98 ± 3.26	0.166
Systolic pressure	135.14 ± 16.77	134.18 ± 13.97	0.560	134.69 ± 14.60	135.80 ± 20.08	0.318	134.76 ± 14.37	135.45 ± 19.27	0.508
Diastolic pressure	78.93 ± 8.83	78.33 ± 7.79	0.488	78.86 ± 8.48	78.87 ± 9.24	0.999	78.78 ± 8.33	79.00 ± 9.28	0.697
Waist circumference	88.77 ± 8.27	88.61 ± 8.10	0.843	88.60 ± 8.10	89.09 ± 8.59	0.369	88.54 ± 8.08	89.07 ± 8.51	0.316
Hip circumference	98.42 ± 7.43	98.68 ± 7.17	0.732	98.39 ± 7.36	98.58 ± 7.50	0.702	98.24 ± 7.32	98.77 ± 7.52	0.256
WHR	0.90 ± 0.05	0.90 ± 0.05	0.478	0.90 ± 0.05	0.90 ± 0.06	0.282	0.90 ± 0.05	0.90 ± 0.05	0.894
CAP	273.89 ± 49.44	264.04 ± 47.78	0.073	275.00 ± 49.79	268.10 ± 48.03	0.055	273.71 ± 50.35	271.60 ± 47.75	0.534
LSM	5.07 ± 1.95	4.85 ± 2.00	0.318	5.14 ± 2.06	4.84 ± 1.68	0.040	5.17 ± 2.05	4.85 ± 1.78	0.019
NFS	−0.74 [−1.44–0.01]	−0.92 [−1.49–−0.08]	0.470	−0.74 [−1.45–−0.05]	−0.76 [−1.46–0.09]	0.849	−0.76 [−1.47–−0.05]	−0.73 [−1.44–0.07]	0.357
LSM ≥ 8.2 kPa, *n* (%)	53 (5.65)	4 (3.48)	0.423	45 (6.18)	12 (3.69)	0.111	39 (6.11)	18 (4.34)	0.250
NFS ≥ −1.455, *n* (%)	597 (63.64)	69 (60.00)	0.295	452 (62.09)	214 (65.85)	0.834	401 (62.85)	265 (63.86)	0.766
TBIL (μmol/L)	15.53 ± 6.90	14.98 ± 4.41	0.406	15.56 ± 7.09	15.28 ± 5.66	0.530	15.57 ± 7.19	15.32 ± 5.80	0.556
TG (mmol/L)	1.59 ± 1.06	1.51 ± 0.92	0.401	1.60 ± 1.05	1.56 ± 1.03	0.579	1.57 ± 1.02	1.60 ± 1.08	0.724
TC (mmol/L)	4.81 ± 1.25	4.89 ± 1.09	0.483	4.80 ± 1.26	4.86 ± 1.15	0.452	4.75 ± 1.25	4.92 ± 1.18	0.031
HDL (mmol/L)	1.15 ± 0.25	1.18 ± 0.27	0.218	1.15 ± 0.25	1.18 ± 0.26	0.074	1.15 ± 0.25	1.16 ± 0.25	0.517
LDL (mmol/L)	3.33 ± 1.13	3.41 ± 1.00	0.511	3.33 ± 1.15	3.37 ± 1.05	0.593	3.28 ± 1.14	3.44 ± 1.08	0.030
ALT (U/L)	19.83 ± 9.31	19.03 ± 9.50	0.389	19.98 ± 9.58	19.21 ± 8.73	0.215	19.89 ± 9.28	19.52 ± 9.41	0.536
AST (U/L)	18.52 ± 7.58	18.31 ± 6.67	0.778	18.63 ± 8.00	18.21 ± 6.14	0.403	18.43 ± 7.62	18.61 ± 7.26	0.711
Hypertension, *n* (%)	722 (77.97)	88 (76.52)	0.850	561 (77.06)	249 (76.62)	0.870	488 (76.49)	322 (77.59)	0.959
T2DM, *n* (%)	398 (42.43)	43 (37.39)	0.293	311 (42.72)	130 (40.00)	0.421	267 (41.85)	174 (41.93)	0.986
Stroke, *n* (%)	146 (15.57)	19 (16.52)	0.804	117 (16.07)	48 (14.77)	0.591	100 (15.67)	65 (15.66)	0.928
Mets, *n* (%)	798 (85.07)	95 (82.61)	0.487	621 (85.30)	272 (83.69)	0.501	539 (84.48)	354 (85.30)	0.718
Lipid lowering agent, *n* (%)	318 (33.90)	37 (32.17)	0.677	242 (33.24)	113 (34.77)	0.649	210 (32.92)	145 (34.94)	0.616

BMI, body mass index; WHR, Waist-to-hip ratio; CAP, controlled attenuated parameter controlled; LSM, liver stiffness measurement; NFS, NAFLD Fibrosis Score; TBIL, total bilirubin; TG, total triglyceride; TC, total cholesterol; HDL, high-density lipoprotein; LDL, low-density lipoprotein; ALT, alanine aminotransferase; AST, aspartate aminotransferase; T2DM, type 2 diabetes; Mets, metabolic syndrome.

**Table 5 biomedicines-11-02416-t005:** Best models to predict NAFLD by generalized multifactor dimensionality reduction (GMDR) ^a^.

GMDR Model	Training Accuracy ^b^	Testing Accuracy ^b^	Sign Test (*p*)	CVC
*rs738491*	0.5468	0.5260	6 (0.3770)	6/10
*rs2073082*, *rs3761472*	0.5571	0.5304	7 (0.1719)	8/10
*rs2073082*, *rs738491* ^a^ *rs3761472*	0.5697	0.5552	9 (0.0107)	10/10

^a^ Whole dataset statistics: Training Balanced Accuracy, 0.5449; Training Accuracy, 0.5449; Training Sensitivity, 0.7300; Training Specificity, 0.3593; Training Odds Ratio, 1.5189 (1.0076, 2.2898); Training χ2 (*p*), 4.0049 (*p* = 0.0454); Training Precision, 0.5327; Training Kappa, 0.0897; Training F-Measure, 0.6159. ^b^ The values’ units are (%).

**Table 6 biomedicines-11-02416-t006:** Logistic regression analysis between each genetic model and risk of NAFLD.

Genotypes	Adjusted OR (95% CI) ^a^	*p*-Value
*rs738491*		
CC	1	
CT + TT	1.532 (1.144, 2.053)	0.004
*rs2073082*, *rs3761472*		
AA, AA	1	
GG + AG, GG + AG	1.809 (1.147, 2.853)	0.011
*rs2073082*, *rs738491*, *rs3761472*		
AA, CC, AA	1	
GG + AG, CT + TT, GG + AG	1.892 (1.196, 2.993)	0.006

^a^ Multivariate logistic regression model adjusted for age, sex and BMI. BMI, body mass index; CI, confidence interval; OR, odds ratio.

## Data Availability

The clinical and histological data used to support the findings of this study are available from the corresponding author upon request.

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
