# Peer review of "SAMM50-rs2073082, -rs738491 and -rs3761472 Interactions Enhancement of Susceptibility to Non-Alcoholic Fatty Liver Disease"

_biomedicines, 2023, doi:10.3390/biomedicines11092416_

Round 1

Reviewer 1 Report (Previous Reviewer 4)

The authors have addressed some but not all of my criticisms.

Page 2 line 12: Ref#13 and #12 have not studied rs2073082, this statement is therefore wrong. 

Page 4: line 6: The red sentence should be part of the next sentence.

Page 4: line 15: dimensionionity? Please give proper reference to the MDR and GMDR analysis tools. 

Table 2 top row: which column is NAFLD and which Non-NAFLD?

Page 7 first sentence after 3.4 Associa...: Which previous studies have shown that the G-allele of rs2073.... increase susceptibility in NAFLD? This is not in ref#12 or #13, and not according to table 3. From table 3 it may be deduced that G-carriership for this SNP increases susceptibility. 

Paragraph 3.4: When considering multiple testing, none of the differences in Table 4 or table S3 can be considered statistically significant. In particular, the remark on CAP values should be removed.

Page 8 line 3 from bottom: To try to convince the reader that the three-locus model is better than the single locus rs2073082 G-carriers vs AA (OR 1.962), here the TA, sign test p-value and CVC of the single locus should be included. 

Page 9 line 2: please substitute allele for carriers (twice). 

Table 5: a training and testing accuracy of ~0.5 % is very low.  

Table 6: The OR of the single locus rs2073082 G-carriers vs AA model is higher than the OR for the three-locus model. Please discuss why one should not stick to the simplest model but bother about the more complex three-locus model. 

Figures 2, 3 and 4 should be removed since they are not explained in the text, do not make sense without commenting and interpreting, and do not add anything to what is already presented. For example, in figure 2 what do the numbers mean on top of the bars in each box? Why are these numbers negative for the controls? In Fig.3, what is on the X-axis and what on the Y-axis, without this info the lines and distances are senseless? How come there is a strong antagonism between rs73... and rs207.. and not redundancy as in figure 4? What do the very low percentages mean on the interaction lines? 

Page 13 the last paragraph preceding 5.Conclusions: Ref#13 has done exactly the same investigation except that rs2143571 was used rather than  rs207... . Therefore, that the current study "was the first investigation" is rather overstated. Moreover, it is very conspicuous that ref#13 is not mentioned a single time in the entire discussion although comparison with their results on the interaction between SAMM50 SNPs and risk for NAFLD is highly warranted and dearly missed. 

.. have showed... should be ... have shown..

Author Response

Reviewer 2 Report (Previous Reviewer 2)

The revised manuscript has been improved and incorporated all the  comments. 

1. All occurrences of the gene symbols should be italicized. 

2. The "Exclusion criteria (n=143)" in Figure 1, should be mentioned outside the box to avoid confusion. As the numbers don't add up otherwise. 

 Minor editing of English language required

Author Response

Reviewer 3 Report (Previous Reviewer 1)

The authors made all changes requested by this reviewer.

Author Response

This manuscript is a resubmission of an earlier submission. The following is a list of the peer review reports and author responses from that submission.

Round 1

Reviewer 1 Report

Genetic mapping to determine which genes are involved with each disease is the future of individualized therapy. Thus, this work addresses two important points, the genes associated with NAFDL and the evaluation of patients over 65 years of age with NAFDL. The text is clear and the study well delineated. There are few suggestions for changes.

 Supplementary Materials: The meaning of abbreviations must be inserted at the foot of all supplementary tables.

Reviewer 2 Report

This manuscript by Zhao et.al. describes the effect of three variants of SAMM50 and their interactions on the development on non-alcoholic fatty liver disease (NAFLD). I feel that the manuscript is suitable for publication in its current form.

The manuscript is written well, with the use of good english language. 

Reviewer 3 Report

The paper describes the investigation of three SNPs in SAMM50 gene in relation to the development of NALFD. The involvement of this gene in the occurrence of the disease is interesting and merits to be investigated. However, the manuscript is not very well written, starting from the English. Often the grammar is not correct and many sentences have no sense. Furthermore, also the genetic analysis is not presented with the correct terminology. Authors make confusion among the concepts of allele and genotype, for example, gene and SNP,  but also among genotype and haplotype. The analyses are many and presented in a confusing way, both regarding tables and figures.

In details my comments:

-        The abstract is not very well written : SNP instead of SNPs, haplotype instead of genotype. The conlusions must be smoothed

-        “Using the χ2 test, genotypes and allele frequencies were computed” The x2 test really is used to calculate differences in alleles o genotypes distributions in different classes (or sample groups), not to compute the frequencies…

-        Table 3: I think the presentation of results creates confusion: if the comparison is C vs T, with an OR=1.216, I intend that the C allele is the risk one, not the T as it is correctly reported in the text. This way to report the comparison needs to be revised in the table and in the text

-        Association of three SAMM50 gene variants with clinical features: the prelude of this paragraph is not correct since the G-allele in rs3761472 is not a risk allele for NALFD susceptibility.

-        Regarding Table 4, the number of variables tested for association is large. Therefore, the Authors should perform the correction for multiple comparisons. Doing so, unfortunately the association with LSM will be lost

-        Why the Authors have not performed a haplotype analysis and compared simply NAFLD and not NAFLD for haplotypes distribution?

-        Analysis of SNP-SNP interactions: this analysis is not very clear to me. If it is specific to analyse SNP’s interactions, why they have tested a single SNP?(the single locus model). Secondary, the p-values of the “single locus” and “rs2073082 and rs3761472” are not significant. So, what is the sense to report the accuracy of these models? What the Authors infer from this not significance?

-         The following sentence  has no sense :Additionally, G-T-G carriers had nearly twice the risk of NAFLD as haplotype A-C-A did”. The haplotype inference is a particular type of genetic analysis in which  it is possible to infer how alleles are transmitted in block.  The Authors instead are referring to carriers of particular genotypes

-         Figure 2: the Figure is not showing a gene-gene interaction. The SNPs are in the same gene

-        I do not understand the meaning of the Fruchterman-Rheingold figure and what it represents

-        Table 6. Again, I do not understand the meaning. Interactions among SNPs and disease? Maybe they refer to SNPs interaction in the development of the disease. Which is the dependent variable? As they wrote it seems that the disease is one of the  independent  variable

-        Discussion section: I do not consider this one a clinical study

-        “We demonstrate for the first time that the best model for predicting NAFLD in elderly individuals included rs2073082, rs738491 and rs3761472 using GMDR analysis.: I think this sentence is too strong,  the predictive model is not a demonstration but a prediction.

-        Discussion: “Therefore, we conducted the current study to explore the clinical features of genetic factors in a cohort …..”  The sentence has no sense

-        “this was the first investigation into the interaction between the three genes (rs2073082, rs738491, and rs3761472).” It seems that the Authors make confusion again among the concept of gene and SNP.

-         In conclusion part : the 0.5552  accuracy could be considered the “most accurate prediction of the predisposition for elderly NAFLD”. What is the range that  measure the accuracy? Is it  between 0 and 1? If so, maybe the accuracy is low. Is there any threshold level to be considered reliable?

Minor comments

-The English needs to be extensively revised .  for example there are many problems of singular/plural terms. But also many sentences with no sense.

Some examples of grammar or syntax mistakes :

-Introduction  section : “Therefore, we conducted this case-control study using to investigate the effects of…..”.

- Statistical analysis “ For comparing continuous variables that were reported as means with standard de-viations (SD) or medians (25th, 75th percentiles), respectively”. Respectively to what???

- Discussion : how their affects

-SAMM50 and its encoded protein: replace with SAMM50 gene and its encoded protein

The manuscript needs to be extensively revised, many sentences have no sense or are written in a very inaccurate way

Reviewer 4 Report

In this case-control design, the effect of three SAMM50 SNPs on the susceptibility to non-alcoholi fatty liver disease (NAFLD) in elderly Chinese people is studied. The authors conclude that two SNPs are significantly associated with NAFLD risk, and two SNPs are related to higher liver stiffness. Furthermore, the authors claim that a three SNP combination model accurately predicts the risk of NAFLD. Unfortunately, I disagree with the latter conclusion. The authors show that the adjusted OR for NAFLD risk is obtained for carriers of the rs 2073082 G allele is 1.962, which is higher than the OR for the three-SNPs model (1.892), thus demonstrating that the rs2073082 SNP alone best predicts NAFLD risk.

In addition, there are serious issues with the study design. First, controls were not matched for cases. There are numerous significant differences between the cases and controls, so do these SNPs predict for NAFLD or for any of the other variables? Secondly, the three SNPs are probably in strong linkage disequilibrium. Therefore, the three SNPs are probably not inherited mutually independent. Thirdly, interaction analysis by GMDR was started from rs738491 instead of the rs2073082 SNP, thereby ignoring the much higher OR of the latter SNP to predict NAFLD risk. Finally, no comparison is made with the predictive potential of SNPs in for example the PNPLA3, GCKR and MBOAT7 genes mentioned in the introduction. It would have been more interesting to test whether there is interaction among the SNPs in these four genes in enhancing the risk of NAFLD.

Minor:

Figure 1: 1280 eligible patients minus 226 patients does not equal 1053 patients included in the study. Why are controls also designated as patients?

Page 3, formula for NFS: is “impaired” FBG different from FBG? …/ diabetes mellitus (Y=1,N=0): does this really mean dividing by zero in case of not having DM?

Page 4, line 9: why is obesity characterized by BMI>25 in stead of the usual 27.5 for Asian people?

Page 4 section 3.1 first line: with only two populations it is not meaningful to give mean age with SD.

Table 1: since only age above 65 was selected, it is probably best to give age distribution in median and IQR. IQR is supposed to be indicated in [ ] brackets, not in ( ).

Page 7 section 3.4 line 1-2: according to table 3, the G-allele of in rs3761472 does NOT increase susceptibility for NAFLD.

Table 4 and supplementary tables S2 till S4 have little added value. None of the statistical differences will remain after correction for multiple testing.

Page 9 section 3.5 lines 3-4: What is testing accuracy of single-locus model with rs2073082? TA is not significant for the single-locus model with rs738491. Similarly, the two-locus model is also not significant.

Figure 2: why do all the bars concerning control subjects have a negative value?

Figure 4: please explain the values given in percentages in the interaction and for each rs. Percentage in the dark blue interaction line is not visible.

Page 12 first paragraph: is it possible that the discrepancies with other studies is explained by the Fibroscan being inferior to biopsies for determining fibrosis?

Page 13 lines 1-2: what does mean that “our research team is listed as the elderly physical examination population"?

Tabel S1: please explain the difference between model 1 and model 2?

minor english corrections needed